# The Role of Women’s Autonomy and Experience of Intimate Partner Violence as a Predictor of Maternal Healthcare Service Utilization in Nepal

**DOI:** 10.3390/ijerph16050895

**Published:** 2019-03-12

**Authors:** Sujan Gautam, Hyoung-Sun Jeong

**Affiliations:** Department of Health Administration, Graduate School, Yonsei University, 1 Yonseidae-gil, Wonju, Gangwon-do 26493, Korea; gautamsujan@gmail.com

**Keywords:** women’s autonomy, intimate partner violence, antenatal care, institutional delivery, maternal health service utilization, Nepal

## Abstract

This study aims to identify the relationship of women’s autonomy and intimate partner violence (IPV) with maternal healthcare service utilization among married women of reproductive age in Nepal. This study used data from the 2016 Nepal Demographic and Health Survey (NDHS), which is a nationally representative sample survey. The association between outcome variables with selected factors were examined by using the Chi-square test (χ^2^), followed by multiple logistic regression. The sample was adjusted for multi-stage sampling design, cluster weight, and sample weight. Of the total sample, 68.4% reported attending sufficient Antenatal care (ANC) visits throughout their pregnancy, while 59.9% reported having a health facility delivery. The factors associated with both, sufficient ANC visits and institutional delivery includes ethnicity, place of residence, household wealth status, and the number of living children. Women who have access to media, and who have intended pregnancy were more likely to have sufficient ANC visits. Exposure to some forms of violence was found to be the barrier for maternal health service utilization. Attending ANC visits enables mothers to make the decision regarding skilled attendance or health facility delivery. Preventing any forms of violence need to be considered as a vital element in interventions aimed at increasing maternal health service utilization.

## 1. Introduction

Globally, approximately 830 women die each day from preventable causes related to pregnancy or childbirth. Among total maternal deaths, almost 99% of deaths occur in developing countries. This shows that the lifetime risk of death due to pregnancy—or childbirth-related complications—is higher in developing countries as compared to developed countries (for example, the lifetime risk of death of 1-year-old women from a maternal cause is; 1 in 180 in developing countries vs. 1 in 4900 in developed countries) [1]. The differences in health and wellbeing of women between developed and developing countries is a growing concern [2].

Maternal health refers to the health of the women during pregnancy, childbirth and the postpartum period, therefore, maternal health service includes antenatal care (ANC), delivery care and postnatal care (PNC). The lives of millions of women of reproductive age can be saved through proper utilization of maternal health care services.

The antenatal care is an entry point for maternal and child healthcare service utilization through which pregnancy risk can be detected and managed and contributes to reducing both the maternal and neonatal mortality [3]. During ANC visits, women are given the health education on the health of mother and child which facilitates them in making a decision for skilled attendance or institutional delivery. It is found that women who utilized ANC services are more likely to utilize skilled attendance at delivery or institutional delivery [4]. Similarly, the second component of maternal health care services, institutional delivery, allows detection and management of risk during labor and childbirth so that effective interventions can be provided by medically trained personnel at a health facility [5].

The government of Nepal is promoting safe motherhood programs through various initiatives such as providing free delivery care and transportation incentives to women delivering in a health facility [6]. The government of Nepal initiated the free delivery care services in 2005 through the ‘Maternal Incentive Scheme’ to increase health facility delivery. In 2009, the scheme was expanded to include ‘free deliveries’ and in the same year, the government of Nepal introduced the “four ANC incentive program” to improve ANC attendance [7]. In 2016, government of Nepal endorsed a ‘Nepal Every Newborn Action Plan’ which sets a vision for the country “in which there are no preventable deaths of newborns or stillbirths, where every pregnancy is wanted, every birth celebrated, and women, babies, and children survive, thrive and reach their full potential” which strengthens the commitment of government and its partners in improving maternal and newborn health care [8]. Still, the proportion of women utilizing maternal health services is low in Nepal. Nationally, sixty-nine percent of women had at least four ANC visits and fifty-seven percent of deliveries were conducted in a health facility [6]. Despite the need and the efforts made by the government of Nepal through different policies to improve access to maternal health care services and reduce maternal mortality, complete maternal health service utilization has been very limited.

A growing body of research shows that numerous factors have been found to be associated with the utilization of maternal healthcare services in Nepal. An array of literature has documented that women residing in rural areas, with lower level of education (herself and her husband), with multiple pregnancies, having unpaid employment status, from lower household wealth status, with low access to media, with lower women’s autonomy, and unintended pregnancy are associated with poor utilization of maternal healthcare services [9,10,11,12,13].

### 1.1. Women’s Autonomy

Women’s autonomy refers to the women’s ability and freedom choose and act independently, and hence includes women’s capability to articulate strategic choices, access to and control over resources, and participate in decision-making [13]. The measures of women’s autonomy such as decision-making autonomy, freedom of movement and economic autonomy are related to women’s status and women’s power and agency in such a way that autonomous women have the power to act independently for her better health [14]. There is evidence of association of higher level of overall women’s autonomy with a higher rate of utilization of maternal health care services [13,15] particularly prenatal and postnatal care services [14]. Couple’s reports of women’s autonomy and health care-use in Nepal showed that given the partner’s agreement about wife’s autonomy, she is more likely to have used health-care services than when spouses disagree about wife’s autonomy [16].

### 1.2. Intimate Partner Violence 

Intimate partner violence (IPV) which refers to “any behavior by an intimate partner that leads to physical, sexual or psychological harm, including physical aggression, sexual coercion, psychological abuse, and controlling behaviors” [17], is found to be associated with poor utilization of maternal health services particularly in developing countries, including Nepal [18,19,20,21]. The poor utilization of maternal health services by the victims of IPV may be due to the lower autonomy on household decision making, decreased freedom of movement and increased economic dependency [18]. This will limit the women’s ability to make a decision regarding herself to choose and receive the appropriate reproductive healthcare services [21]. Also, there is evidence of association of maternal experience of violence with an increased risk of malnutrition among mother and children [22]. Although many studies are done to assess the factors related to utilization of maternal health services in Nepal [23,24,25], little is known about the link between women’s autonomy, different forms of IPV and partner control behavior with the utilization of maternal healthcare services. Therefore, this study aims to identify the relationship of women’s autonomy and exposure to different forms of IPV with maternal healthcare service utilization among Nepalese women of reproductive age.

## 2. Materials and Methods

### 2.1. Study Area, Study Design, Data, and Sampling

Nepal is a landlocked developing country in South-Asia. The 2015 Constitution of Nepal establishes the country as a federal secular parliamentary republic divided into seven provinces. We used the data from the 2016 Nepal Demographic and Health Survey (NDHS), a nationally representative sample survey. The 2016 NDHS survey conducted by New ERA under the support of the Ministry of Health of Nepal is the fifth comprehensive cross-sectional survey of its kind conducted as a part of Demographic and Health Surveys (DHS) [6]. The dataset of this survey is publicly available from ‘The DHS Program’ website [26]. Details of the questionnaires and procedures used in the survey can be found on the website and survey report [6,26]. Briefly, the 2016 NDHS used a multi-stage cluster sampling design to collect the data. Each province was stratified into rural and urban area yielding 14 sampling strata. The stratified sample was selected in two stages in rural areas, and in three stages in the urban area. The primary sampling units (PSUs) for 2016 NDHS is a ward in rural areas, whereas, in urban areas, one enumeration areas (EAs) was selected from each PSUs. A stratified, multistage cluster sample of 383 PSUs was constructed (184 in urban and 199 in rural areas). In the final stage, 30 households per cluster (fixed) were selected using systematic sample selection. In this way, 12,862 women aged 15–49 years successfully completed the survey with a response rate of 98%. Inclusion criteria for our sample included only those currently married women who were selected for the domestic violence module and had a live birth in the past five years of the survey (not fit in this criteria are excluded). Maternal health service utilization was assessed for the most recent birth (if a woman had more than one birth within past five years) to minimize the recall bias [24], which yielded a sample size of 1361 (weighted value). The domestic violence module was applied to the sub-sample of households selected for the men’s survey and administered to only one eligible woman per households selected randomly.

### 2.2. Measurement of Variables

#### 2.2.1. Outcome Variables

The outcome variable used for this study was maternal health service utilization for recent birth in the past five years. Although maternal health care has various components, we used two items, viz.: (i) the sufficient amount of ANC visits throughout the pregnancy, and (ii) delivery in a health facility (institutional delivery) as proxy outcome variables to measure maternal health service utilization, as shown in Table 1.

#### 2.2.2. Explanatory Variables

This study used three groups of explanatory variables: traditional socio-demographic indicators of women; women’s autonomy related; and, violence related factors, which are listed in Table 2. These variables were chosen based on previous studies on maternal health service utilization done nationally and internationally [13,15,21,23,25,29,30,31,32,33]. The socio-demographic variables include the age of women, ethnicity, place of residence, household wealth status, education differences with husband, and the number of living children. NDHS 2016 evaluated household wealth index using scores derived from principal component analysis of various household possessions, and amenities [34]. The scores were divided into five quintiles each comprising 20% of the population and ranked as: poorer; poor; middle; rich; and, richest. For our analysis, we re-categorized wealth into three categories [24,25]. Women’s autonomy was assessed through five different areas: Women’s employment; access to media; household decision making power; attitudes towards partner violence; and pregnancy intentions. Women’s who had a paid job (in cash); had access to media at least once a week; and, having intended pregnancy when she became pregnant were considered to have high autonomy. Exposure to mass media is considered as the main source of health seeking information in low-and-middle-income countries [35]. Therefore, women who have access to media at least once a week are considered to possess high autonomy regarding seeking health service utilization. Women’s attitude towards partner violence is assumed to measure the gender role norms that justifies men’s control over women. The assumption is that women who have an attitude of not accepting gender inequalities norms of marital violence were considered to have high autonomy [15]. Violence related factors include marital control behavior from partner; physical IPV; sexual IPV; emotional IPV; and any IPV (physical, sexual, and/or emotional IPV). Shortened and modified version of the Conflict Tactics Scale (CTS-2) was used to measure IPV [36], and the World Health Organization (WHO) ethical guidelines were followed while collecting information on violence module [37]. The Cronbach’s alpha for the 7-item physical IPV was 0.876; sexual IPV was 0.899; emotional IPV was 0.835; and any IPV (13-item scale) was 0.903.

### 2.3. Statistical Analysis

This study includes two outcome variables; (i) Sufficient ANC visits; and, (ii) Institutional delivery, used as a proxy measure of the maternal health service utilization. Descriptive statistics were used to describe the characteristics of the respondents and to report the rate of utilization of maternal health service. Then, the chi-square (χ^2^) test of independence was used to examine the relationship between independent and outcome variables. The two-tailed level of significance for all analyses was set at *p* < 0.05, and the variables with a level of significance *p* < 0.05 in chi-square tests were included in multivariate analysis. Prior to the multivariate regression analysis, all the variables were checked for multicollinearity by examining the variance inflation factor (VIF) and found that VIF <2.0. To conceptualize the analysis, explanatory variables were categorized into different groups that could affect the utilization of maternal health services. Logistic regression models were fitted using hierarchical modeling strategy as done in the previous studies using the DHS data [13,24,38]. Three adjusted models were created to analyze the factors associated with sufficient ANC visits. For institutional delivery, we considered sufficient ANC visits as an independent variable [23,30]. In multivariate regression analysis, model 1 included traditional socio-demographic factors, model 2 included factors of model 1 and women’s autonomy related factors, and model 3 consists of factors of model 2 and violence related factors. The fourth model (only for institutional delivery) consists of factors of model 3 and sufficient ANC visits. For both the outcomes, an extra alternative model was created to see if the composite variable of IPV has any change in effect in the final model. An adjusted odds ratio (AOR), 95% confidence intervals (CIs), and *p*-value were reported. All the statistical analyses were performed using complex sample analysis procedure to allow for adjustment for the sampling weight, stratification, cluster sampling design, and the calculation of standard errors of the large survey data [39]. Such sampling weights were provided by the 2016 NDHS survey to make the data nationally reproducible. IBM SPSS version 25.0 was used for all analyses.

### 2.4. Ethical Approval

The data collection protocol of 2016 NDHS survey was reviewed and approved by the ICF International Review Board and the Nepal Research Health Council (NHRC). Therefore, independent ethical approval was not obligatory. To ensure the privacy in the survey, the questions on violence were administered to only one eligible woman per household selected randomly.

## 3. Results

### 3.1. Basic Characteristics of the Study Population, NDHS 2016

The basic characteristics of the 1361 study participants are shown in Table 3.

#### 3.1.1. Socio-Demographic Characteristics

Of the total 1361 respondents, almost half (48.8%) were in the 25–34-year-old age group; more than one third (34%) had Janajati (Indigenous) ethnicity; more than half (54%) were from an urban area; and 42.6% were poor. Only about 14% of women were better educated than their husbands, and the majority (71.6%) had two or fewer children.

#### 3.1.2. Women’s Autonomy Related Characteristics

As far as women’s autonomy related factors are concerned, most of the women’s were relatively unemployed (37.5%); more than half (55.7%) were exposed to any type of media at least once a week; about one third (32.7%) had high autonomy in household decision making; about 29% had an attitude of accepting violence (wife-beating) by their husbands for any single or several reasons; and only 9% of them reported unintended pregnancy when they were pregnant.

#### 3.1.3. Intimate Partner Violence Related Characteristics

The most common act of violence reported by the respondents was partner controlling behavior (35.4%) showed by their husbands. The second form of violence reported by the women was physical IPV (21.6%), followed by emotional, and sexual IPV which was 12.4% and 7.4%, respectively. The proportion of women who have ever experienced any form of IPV was 25.3% which showed that some women experienced multiple forms of IPV.

### 3.2. Rate of Utilization of Maternal Healthcare Service in Nepal

Table 3 shows the rate of utilization of maternal healthcare services, measured in terms of: sufficient ANC visits throughout the pregnancy, and, institutional delivery the women had for the most recent birth in the past five years, by explanatory variables.

#### 3.2.1. Sufficinet ANC Visits

Of the 1361 women, more than two-thirds of the respondents (68.4%; 95% CI: 64.9–71.8%) had attended sufficient ANC visits (≥4 visits) throughout their pregnancy. The proportion of sufficient ANC visits was found to be decreased with the increase in age (73.2% among women aged 15–24 years vs. 54.7% among women aged 35–49 years). Women with Brahmin/Chhetri ethnicity (78.9%), residing in an urban area (75.1%), belonging to the rich family (80.9%), the couple being equally educated (81.3%), and having two or fewer children (76.9%) were associated with higher utilization of sufficient ANC visits. The utilization of sufficient ANC visits was found almost similar among employment categories. Women who are exposed to any media (78.1%), having high autonomy in household decision making (71.7%), and have an attitude of rejecting partner violence (69.9%) had high utilization of sufficient ANC visits. Higher utilization of sufficient ANC visits was associated with pregnancy being intended (71.2%). The proportion of utilization of sufficient ANC visits was relatively less among women who experienced violence. For example, only 58.8% of women who experienced any forms of IPV had sufficient ANC visits as compared to women who did not experience (71.7%). In the bivariate analysis (chi-square test), all the socio-demographic characteristics, some women’s autonomy related factors such as exposure to media and pregnancy intentions, and all the violence related factors were found to have significant differences on the utilization of sufficient ANC visits (Table 3).

#### 3.2.2. Institutional Delivery

Of the 1361 women, almost six in ten women (59.9%; 95% CI: 56.1–63.5%) had delivery in a health facility. The higher rate of institutional delivery was associated with lower maternal age (66.3% among women aged 15–24 years vs. 51.4% among women aged 35–49 years). The higher proportion of institutional delivery was found among the women with Brahmin/Chhetri ethnicity (68.6%), from an urban area (69.0%), belonging to the rich family (79.4%), having two or fewer children (70.1%) and couples having equal level of education (71.5%). Women who were employed but not for cash (49.7%), having no access to media (47.5%), having no autonomy in household decision making (55.7%) had a low rate of institutional delivery whereas, women who had an attitude towards rejecting partner violence (61.4%), and whose pregnancy was intended (61.7%) was associated with high rate of institutional delivery. The rate of delivery in a health facility was lower among women who had been the victims of violence. The higher proportion of institutional delivery was found among the women who had sufficient ANC visits (71.7%). The Chi-square test showed that all the socio-demographic factors, few women’s autonomy related factors such as women’s employment status, exposure to media and pregnancy intentions, and some violence-related factors such as physical IPV and any IPV had significant differences on institutional delivery. In addition, when the utilization of sufficient ANC visits was considered as independent variables, it was significantly associated with institutional delivery (Table 3).

### 3.3. Multivariate Analysis

#### 3.3.1. Factors Associated with the Sufficient ANC Visits

Table 4 shows the complex sample logistic regression analysis of the factors associated sufficient amount of ANC visits among women in Nepal. All the socio-demographic variables except women’s age were significantly associated with the utilization of sufficient ANC visits (Model 1). The women’s autonomy-related factors were significantly associated with utilization of sufficient ANC visits after adjusting for socio-demographic factors (Model 2). When different forms of violence related indicators were added in model 2, all the significant variables in model 2 remained significant, and of all the violence related factors, only partner control behavior had a significant effect on utilization of sufficient ANC visits (Model 3). In the adjusted model (Model 3), women with Brahmin/Chhetri ethnicity (AOR: 2.34, CI: 1.42–3.84), couples being equally educated (AOR: 2.31, CI: 1.33–4.02), having two or fewer children (AOR: 2.10, CI: 1.44–3.08), having access to media (AOR: 1.50, CI: 1.07–2.09), and pregnancy being intended (AOR: 1.96, CI: 1.13–3.40) were significantly associated with higher utilization of sufficient ANC visits. However, women residing in the rural area (AOR: 0.69, CI: 0.50–0.96), belonging to the poor family (AOR: 0.47, CI: 0.31–0.70), and having witnessed 1–2 partner control behavior (AOR: 0.64, CI: 0.45–0.92) were significantly associated with lower utilization of sufficient ANC visits. Women’s current age, different forms of IPV such as physical IPV, sexual IPV, and emotional IPV did not have any significant association with utilization of sufficient ANC visits.

In an alternative multivariate regression model of sufficient ANC visits (Alternative model), when different forms of IPV was replaced with a single aggregate variable of IPV (any IPV) in model 3, the yielded result was similar to that of model 3, and that single aggregate variable of IPV (any IPV) did not have any significant association with utilization of sufficient ANC visits. There were no any changes in significance and direction of associations among other variables in the alternative model compared to model 3 (Table 4).

#### 3.3.2. Factors Associated with Delivery in a Health Facility (Institutional Delivery)

Table 5 shows the complex sample logistic regression analysis of the factors associated with institutional delivery among women in Nepal. In model 1, all the socio-demographic variables except women’s age were significantly associated with utilization of institutional delivery. Of all women’s autonomy related factors added in model 1, only women’s employment was found to be significantly associated with utilization of institutional delivery after adjusting for socio-demographic factors (Model 2). When violence related factor (women’s experience of physical IPV) was added in model 2, all the variables significant at model 2 remained significant, and physical IPV showed a significant association with the use of institutional delivery (Model 3). Considering utilization of sufficient ANC visits as an independent variable for predicting institutional delivery, the association was found to be statistically significant, and all the variables significant in model 3 except education differences remained significant (Model 4). In the adjusted model (Model 4), women with Brahmin/Chhetri ethnicity (AOR: 2.26, CI: 1.37–3.72), being better educated than their husbands (AOR: 1.95, CI: 1.04–3.65), having two or less children (AOR: 2.51, CI: 1.70–3.69), and utilizing sufficient ANC visits (AOR: 3.06, CI: 2.20–4.26) were significantly associated with higher rate of institutional delivery. However, women residing in rural areas (AOR: 0.67, CI: 0.47–0.96), belonging to poor family (AOR: 0.26, CI: 0.17–0.39), employed but not for cash (AOR: 0.53, CI: 0.38–0.75), and victims of physical IPV (AOR: 0.67, CI: 0.47–0.96) were significantly associated with lower use of institutional delivery. Other factors such as women’s current age, access to media, and pregnancy intentions did not have any significant association with utilization of institutional delivery.

In an alternative multivariate regression model of institutional delivery (Alternative model), the violence related variable (physical IPV) was replaced by a single aggregate variable of IPV (any IPV). The result obtained in an alternative model was comparable to model 4, and no changes in the significance and direction of association among these variables were found, except for the aggregate variable for violence (any IPV) which did not show any significant association with the use of institutional delivery (Table 5).

## 4. Discussion

This study aimed to identify the effect of women’s autonomy and IPV victimization on maternal health service utilization among women aged 15–49 years who had a birth in the past five years. We considered only two components, (i) sufficient ANC visits and (ii) institutional delivery as a proxy outcome measure to evaluate maternal health service utilization. The issues of maternal health service utilization and exposure to IPV is most important to explore within the context of prevalent interpersonal violence which is troubling several countries [40]. Exposure to different forms of IPV (physical, sexual, and emotional) is the only form of interpersonal violence which is measured in DHS using household questionnaire. It is the basic assumption that if women have autonomy in household decision making, if women are employed, have control over financial resources, have access to media, and have an attitude opposing towards partner violence, they are more likely to utilize maternal healthcare service, whereas women who are exposed to any form of IPV are less likely to utilize maternal healthcare service.

More than two thirds of the women (68.4%) had sufficient ANC visits during their most recent pregnancy and about three out of five women (59.9%) had delivery done in the health facility. These rates were higher when compared to the national average of 2011 [41] and some other developing nations [15,31]. Higher rates of sufficient ANC utilization may reflect higher use of maternal health service in Nepal, however, it does not indicate the quality of care received during those visits. The lower rate of institutional delivery compared to sufficient ANC visits can be attributed to the traditional Nepalese society where childbirth used to take place at home, and many women still may hold the same view thinking that institutional delivery is unnecessary [23].

This study found out that women residing in the rural area, belonging to a poor family, were less likely utilize maternal health care services, which is similar to the findings of the previous study [11]. The cost associated with care can be the barrier for poor women to utilize maternal healthcare services [7,42]. Rural residency could be attributed to fewer health facilities and lack of a well-functioning transportation system to reach the health facility, especially in developing countries. A qualitative study done in Indonesia showed that the physical distance to reach health facilities, and financial difficulty were the barriers to accessing antenatal and postnatal care services [43]. In contrast to the findings of this study, a multilevel analysis done in Colombia did not find any significant association between health service use and place of residence [44].

Women with a lower number of living children (two or fewer children) were found to utilize maternal health services more compared to women with a higher number of children. Other studies found a similar result of less utilization of maternal healthcare services with the increase in the number of births [7,45]. With the higher number of births, women may feel experienced and are less likely to seek maternal healthcare services [46] whereas, the perceived risk associated with initial pregnancies could be the reason for high utilization of maternal health services [7,30].

Utilization of sufficient ANC visits was found more if the couples are equally educated, while women with a better education than her partner was more likely to have health facility delivery. In another study, women who were less educated than her partner had completely opposite relationship with access to skilled antenatal care in Nepal and India such that they were less likely in Nepal and more likely in India to have access to skilled antenatal care [47]. This indicates the variation and complexities of gender-related norms within different social contexts. Higher utilization of maternal health service associated with women being equally or more educated than her husband reflects the uplifting status within their relationships.

This study did not show any relationships between women household decision making, and women’s attitude towards partner violence with both the component of maternal health service utilization. Consistent with our findings, a study done among young married women in Nepal did not show any significant associations between women’s autonomy in household decision making and institutional delivery [30]. However, some studies showed women who had autonomy in household decision making were more likely to utilize maternal health services and asserts in their study that utilization of maternal health services is influenced by women’s roles in the decision-making process which enables them to lessen their reproductive behavior risks [13]. Studies done in some countries showed that attitudes towards wife-beating were found to be significantly associated with different maternal health care services such that women who had opposing attitudes towards wife beating were more likely to use maternal health care services [15,33]. A study done in four countries including Nepal also showed the opposing attitudes towards partner violence affected the utilization of maternal healthcare services [47].

In this study, women’s employment did not have any associations with sufficient ANC visits even in bivariate analysis however, the relationship was significant for institutional delivery even in an adjusted logistic regression model. Women’s who were employed but not for cash were less likely to have health facility delivery. A study done in Ethiopia showed that women’s employment was significantly associated with institutional delivery but not for sufficient ANC utilization, which is similar to our study findings [33]. A study done in Nepal revealed that women’s employment status did not translate clearly into the higher utilization of maternal healthcare services and found out that women who work but did not have control over their earnings are less likely to utilize maternal healthcare services [48]. The women who are employed for cash would have greater influence over their healthcare as compared to the women who are unemployed or employed but not for cash. And the lower utilization of health facility delivery among women who are unemployed or employed but not for cash may reflect their economic dependence on their partners for health service use.

In our study, women who were exposed to media were significantly more likely to have sufficient ANC visits, however, access to media did not have any significant association with institutional delivery. Previous studies done in Nepal showed that women exposed to various types of mass media were more likely to attend and receive ANC care [9,10,49]. In the developing country like Nepal, mass media such as radio, TV, and newspaper are very effective in playing an important role in the dissemination of health education and information and can influence to mothers in utilizing healthcare services in improving public health.

This study showed that intended pregnancy was significantly associated with sufficient ANC utilization but not for institutional delivery. Consistent with this finding, the results from various countries showed that women having unintended pregnancies were less likely to have adequate prenatal care service utilization [10,50,51]. Undesired pregnancy can be the predisposing factor for delayed or insufficient prenatal care service utilization.

Women who reported marital control from their partner were less likely to use sufficient ANC visits during their last pregnancy. Similar was the result obtained in a study done in Ethiopia where they claimed that partner’s control may lead to lower the women’s confidence in seeking healthcare services [31]. This study did not find any significant associations of exposure to any forms of IPV with sufficient ANC attendance. The possible explanation for this may be the existence of social norms for seeking health service during pregnancy such that exposure to IPV is not a barrier for women in Nepal for utilizing sufficient ANC visits. However, we found that physical IPV was associated with lower utilization of institutional delivery. Findings from other developing countries showed that physical IPV was associated with both insufficient ANC visits and institutional delivery [31]. Other studies done in different countries found out that exposure to IPV was significantly associated with insufficient ANC utilization [19,20,21,40,52]. Women’s who were economically dependent on their husband were more likely to experience physical IPV [53]. Therefore, it can be argued that poor utilization of ANC services by the women who experienced IPV may be due to less autonomy on decision-making and freedom of movement, and increased economic dependency and this could result in a delay of health-seeking behavior for ANC services [18].

This study revealed the significant relationship between ANC attendance and institutional delivery. Women who attended sufficient ANC visits were more likely to have institutional delivery. This result is consistent with other studies done in developing countries of Africa and South-Asia including Nepal [4,7,12,23,25,33]. Women who have sufficient ANC visits can be regarded as more health conscious than women with insufficient ANC visits. Also, ANC visits provide the platform for passing health education and information regarding the health of mother and child and facilitates pregnant women in decision-making regarding skilled attendance at delivery or institutional delivery. An intention to deliver at the health facility can largely be enhanced through extensive counseling on the benefits and safety of health facility delivery [23].

### Strength and Limitations of the Study

The study is based on the data from a large national population-based survey. Since this study considered women’s experience of any forms of violence as one of the predictor variables, the sample was weighted using domestic violence weight provided by the study. This study also weighted the data to adjust for the multistage sampling method used in the survey to make the findings nationally representative. The findings of this study can provide policy makers to design interventions to address maternal healthcare needs.

Nevertheless, this study has some limitations, therefore the results must be construed with caution. The cross-sectional nature of the study does not allow to establish cause-and-effect relationships. Domestic violence, being a sensitive issue and stigma attached to it, there can be some reporting bias. This study did not include some factors which were known to influence maternal health service utilization (for example; distance to health facility, medically trained/untrained service provider during ANC visits or delivery). Also, the postnatal care component of maternal health service utilization was not considered in this study. Since NDHS 2016 collected data from the women who had a birth in the past five years, there can be recall bias. Moreover, this study followed the old guidelines of four ANC visits as sufficient ANC visits but, new recommendations from WHO focuses on a minimum of eight ANC visits for a positive pregnancy experience [54].

## 5. Conclusions

The current study found a relatively low rate of utilization of maternal health services by Nepalese women of reproductive age. The rate of utilizing sufficient ANC visits was found to be more compared to institutional delivery. Attending ANC visits enables mothers to make a decision regarding skilled attendance or health facility delivery. Therefore, it is essential to encourage pregnant women to use the sufficient number of ANC visits. Our study findings revealed that the wide-range of socio-demographic factors was associated with the utilization of maternal healthcare services. Increasing awareness and improving access and coverage to services especially for the rural, poor and uneducated women may increase the rate of utilization of maternal health services as these particular groups were less to use maternal health services. Additionally, the health interventions should give priority to multiparous women as these women were found reluctant to utilize maternal healthcare service. One in four women who had a birth in the past five years reported having experienced any forms of IPV in their lifetime. Marital control from the partner was associated with insufficient ANC visits whereas, exposure to physical IPV was found to be the barrier for institutional delivery. Therefore, in addition to a wide range of socio-demographic factors, preventing any forms of violence need to be considered as a vital element in interventions aimed at increasing maternal health service utilization. The government can focus on policy formulations regarding educational and income-generating programs for women to enhance their status which can effectively increase maternal health service utilization and improve women’s health. In other words, the rate of utilization of maternal health services can be increased by empowering women educationally, socially and economically. Although this study provides a better understanding of women’s autonomy and different forms of IPV with the utilization of maternal healthcare services, further studies, especially more intensive longitudinal studies are desired to further understand the causal relationship of women’s autonomy, and IPV with maternal health service utilization.

## Figures and Tables

**Table 1 ijerph-16-00895-t001:** Measurement of outcome variables.

Outcome Variables	Measurements
Sufficient Antenatal Care (ANC) visits	This factor was assessed in the study as a dichotomous variable created from the continuous measure of the number of ANC visits during pregnancy, grouped as Sufficient ANC visits (four or more); and Insufficient ANC visits (less than four, which also includes no ANC visits). Four or more ANC visits was considered as Sufficient ANC visits based on the standard recommended by the World Health Organization (WHO) [27,28].
Delivery in a health facility (institutional delivery)	Delivery in a health facility (institutional delivery was assessed as a dichotomous variable categorized as: Institutional delivery (when the delivery was done at public, private or non-governmental health facilities); and No institutional delivery (when the delivery took place at home and others). This binary outcome variable was chosen instead of ‘skilled assistance during delivery’ to stress the utilization of institutional delivery services [23,25].

**Table 2 ijerph-16-00895-t002:** Measurement of Explanatory Variables.

Explanatory Variable	Definition and Measurement
Age group (in years)	Self-reported age of women at the time of the survey, grouped into: 15–24 years; 25–34 years; and 35–49 years.
Ethnicity	Self-reported ethnic affiliation of respondents grouped into: Brahmin/Chhetri (Hill Brahmin/Chhetri, Terai Brahmin/Chhetri); Janajati (Newar, Hill/Terai Janajati); Dalit (Hill/Terai Dalit); and Other castes (all other ethnicities).
Place of residence	Types of place of residence: Urban or Rural
Household wealth status	A composite index of household possessions, assets, and amenities, derived using principal component analysis, grouped as: Poor (Poorest and Poorer); Middle; and Rich (Richer and Richest).
Education differences	The highest level of education attained by respondent and her husband/partner collected as No Education, Primary, Secondary, and Higher redefined as education differences between respondent and her husband, grouped into: Both uneducated; Both equally educated; Husband better educated; and, Wife better educated.
No of living children	The number of living children grouped as: 2 or fewer children; and, 3 or more children
Employment status	Employment status classified according to the self-report of types of earning from respondent’s work, grouped into: Unemloyed; Employed but not for cash (paid as In-kind only); Employed for cash (paid as cash only or/and cash and in-kind).
Exposure to any media (at least once a week)	A composite variable derived from the frequency of access to newspaper/magazine, radio and/or television, at least once a week, grouped as: No exposure; Exposure (exposure to any media at least once a week)
Decision-making autonomy	A composite variable measured from women’s participation (alone or with husband) in making three household decisions (access to health care; large household purchases; and freedom to visit families and relatives), grouped into, low autonomy (No participation in any decision making); Medium autonomy (Participation in 1–2 decision making); High autonomy (Participation in all 3-decision making)
Attitudes towards partner violence	A composite variable reflecting women’s attitudes towards wife beating by their husband for each of the following five reasons (goes out without telling her husband; neglects the children; argues with husband; refuses to have sex with husband; and burns the food), grouped as: Accepts violence fully (wife beating justified for 3–5 reasons); Accepts violence partially (beating justified for 1–2 reasons); Rejects violence (Beating not justified for any reasons)
Pregnancy intentions	A dichotomous variable reflecting the women’s pregnancy intentions for the last birth, grouped as: Unintended (pregnancy not wanted at all); and Intended (pregnancy wanted at the time of conception or later).
Partner control behavior	A composite variable reflecting respondent self-reporting of five controlling behavior displayed by the husband/partner (is jealous if she talks to other men; accuses respondent of being unfaithfulness; does not permit respondent to meet female friends; tries to limit respondent’s contact with family; insists on knowing where respondent was), grouped into: No behavior displayed; 1–2 behavior displayed; 3 or more behavior displayed
Physical IPV	A composite binary variable measured by asking women’s if their husband ever did any of the seven following acts against her: pushed, shook or thrown something at her; slapped her; twisted arm or pulled her hair; punched her with fist or something that could hurt her; kicked, dragged or beat her; tried to choke or burn her on purpose; and, threatened or attacked her with any weapon such as knife, gun or any other weapon, grouped as: Yes (experiencing at least one of these seven acts); and No (not experiencing at all).
Sexual IPV	A composite binary variable measured based on three questions by asking women whether their husband ever: physically forced to have unwanted sexual relationships with him; physically forced to perform any other unwanted sexual acts; and, forced with threats and any other way to performs unwanted sexual acts, grouped as: Yes (experiencing any of these three acts): and No (not experiencing at all).
Emotional IPV	A composite binary variable created from the women’s responses to three questions by asking them whether their husband ever: humiliated her in front of others; threatened to hurt or harm her or someone close to her; and, insulted or made her feel bad about herself, grouped as: Yes (if women experienced any of these three acts); and No (not experienced at all).
Any IPV	A composite dichotomous summary measure created from 13 questions (physical IPV: 7, sexual IPV: 3, and emotional IPV: 3) to capture the women’s ever experience of any IPV (physical, sexual, and/or emotional), grouped as: Yes (‘yes’ response to any of these 13 questions); and No (‘no’ response to all of the 13 questions).

**Table 3 ijerph-16-00895-t003:** Basic characteristics and rate (%) of utilization of maternal health service by socio-demographic, women’s autonomy and violence related characteristics, NDHS 2016 (*n* = 1361) ^a^.

Characteristics	Total ^#^ *n* (%)	Maternal Health Service Utilization	
		Sufficient ANC Visits ^#^ *n* (%)	Institutional Delivery ^#^ *n* (%)
**Socio-demographic characteristics**			
Age group (in years)		*p* = 0.005 *	*p* = 0.006 *
15–24	552 (40.6)	404 (73.2)	366 (66.3)
25–34	664 (48.8)	448 (67.5)	374 (56.4)
35–49	145 (10.6)	79 (54.7)	74 (51.4)
Caste/Ethnicity		*p* < 0.001 *	*p* = 0.001 *
Brahmin/Chhetri	381 (28.0)	301 (78.9)	261 (68.6)
Janajati (Indigenous)	462 (34.0)	325 (70.4)	288 (62.4)
Dalit	200 (14.7)	129 (64.7)	108 (53.9)
Other castes	317 (23.3)	176 (55.4)	157 (49.4)
Place of residence		*p* < 0.001 *	*p* < 0.001 *
Rural	632 (46.4)	384 (60.8)	312 (49.4)
Urban	729 (53.6)	547 (75.1)	502 (69.0)
Household wealth status		*p* < 0.001 *	*p* < 0.001 *
Poor	580 (42.6)	344 (59.3)	242 (41.7)
Middle	283 (20.8)	185 (65.3)	177 (62.7)
Rich	497 (36.6)	402 (80.9)	395 (79.4)
Education differences		*p* < 0.001 *	*p* < 0.001 *
Both uneducated	148 (10.9)	61 (41.4)	47 (31.7)
Both equally educated	512 (37.6)	416 (81.3)	366 (71.5)
Husband better educated	510 (37.5)	313 (61.3)	270 (52.9)
Wife better educated	190 (14.0)	141 (74.0)	131 (69.0)
No of living children		*p* < 0.001 *	*p* < 0.001 *
2 or fewer children	975 (71.6)	749 (76.9)	683 (70.1)
3 or more children	386 (28.4)	182 (47.1)	131 (34.1)
**Women’s autonomy related**			
Employment status		*p* = 0.826	*p* < 0.001 *
Not employed	510 (37.5)	343 (67.2)	334 (65.5)
Employed but not for cash	483 (35.5)	336 (69.5)	240 (49.7)
Employed for cash	367 (27.0)	253 (68.8)	240 (65.3)
Exposure to media		*p* < 0.001 *	*p* < 0.001 *
No exposure	603 (44.3)	339 (56.3)	286 (47.5)
Exposure	758 (55.7)	592 (78.1)	528 (69.7)
Decision-making autonomy		*p* = 0.331	*p* = 0.182
No autonomy	445 (32.7)	293 (65.8)	248 (55.7)
Medium autonomy	470 (34.6)	319 (67.9)	294 (62.6)
High autonomy	445 (32.7)	319 (71.7)	272 (61.1)
Attitude towards partner violence		*p* = 0.349	*p* = 0.316
Accepts violence (fully)	35 (2.6)	23 (66.2)	21 (58.9)
Accepts violence (partially)	361 (26.6)	234 (64.7)	202 (56.0)
Rejects violence	964 (70.9)	674 (69.9)	592 (61.4)
Pregnancy intentions		*p* <0.001 *	*p* < 0.001 *
Unintended	122 (9.0)	50 (40.8)	50 (41.4)
Intended	751 (91.0)	882 (71.2)	764 (61.7)
**Violence related**			
Partner control behavior		*p* = 0.016 *	*p* = 0.505
No behavior	879 (64.6)	631 (71.8)	537 (61.1)
1–2 behavior	354 (26.0)	221 (62.3)	206 (58.2)
3–5 behavior	127 (9.4)	80 (62.4)	71 (55.8)
Physical IPV		*p* < 0.001 *	*p* < 0.001 *
No	1067 (78.4)	763 (71.5)	678 (63.5)
Yes	293 (21.6)	168 (57.3)	136 (46.5)
Sexual IPV		*p* = 0.005 *	*p* = 0.420
No	1260 (92.6)	875 (69.5)	759 (60.2)
Yes	101 (7.4)	56 (55.6)	56 (55.5)
Emotional IPV		*p* = 0.005 *	*p* = 0.091
No	1193 (87.6)	835 (70.0)	728 (61.1)
Yes	168 (12.4)	96 (57.1)	86 (51.3)
Any IPV		*p* < 0.001 *	*p* = 0.002 *
No	1016 (74.7)	729 (71.7)	641 (63.1)
Yes	345 (25.3)	203 (58.8)	173 (50.3)
**Maternal Health Service Use**			
Sufficient ANC visits			*p* < 0.001 *
No (less than 4 visits)	429 (31.6)		146 (34.1)
Yes (4 or more visits)	931 (68.4)		668 (71.7)
Institutional delivery			
No	546 (40.1)		
Yes	815 (59.9)		

*n*: number. %: percentage. ^a^ Weighted sample size. ^#^ The number and percentage are adjusted for multi-stage sampling, cluster weight, and sample weight. P refers to a *p*-value of the Chi-square (χ^2^) test. * refers to a statistically significant association (*p* < 0.05) in the χ^2^ test. IPV: Intimate Partner Violence. ANC: Antenatal Care.

**Table 4 ijerph-16-00895-t004:** Factors associated with sufficient ANC visits among women in Nepal, NDHS 2016.

Characteristics	Sufficient ANC Visits; AOR (95% CI)			
	Model 1	Model 2	Model 3	Alternative Model
**Socio-demographic characteristics**				
Age group (years)	*p* = 0.769	*p* = 0.966	*p* = 0.936	*P* = 0.945
15–24	1	1	1	1
25–34	1.08 (0.74–1.57)	1.04 (0.72–1.51)	1.06 (0.74–1.53)	1.06 (0.73–1.53)
35–49	0.93 (0.55–1.58)	1.02 (0.59–1.74)	1.04 (0.60–1.80)	1.03 (0.60–1.79)
Caste/Ethnicity	*p* < 0.001	*p* = 0.001	*p* = 0.003	*p* = 0.003
Other castes	1	1	1	1
Brahmin/Chhetri	2.72 (1.67–4.41)	2.55 (1.55–4.20)	2.34 (1.42–3.84)	2.34 (1.42–3.88)
Janajati (Indigenous)	1.48 (0.89–2.46)	1.40 (0.84–2.36)	1.33 (0.79–2.23)	1.33 (0.80–2.21)
Dalit	1.79 (1.04–3.10)	1.75 (0.99–3.08)	1.70 (0.95–3.04)	1.71 (0.96–3.03)
Place of residence	*p* = 0.034	*p* = 0.034	*p* = 0.028	*p* = 0.026
Urban	1	1	1	1
Rural	0.70 (0.50–0.97)	0.70 (0.50–0.97)	0.69 (0.50–0.96)	0.69 (0.50–0.95)
Household wealth status	*p* < 0.001	*p* = 0.002	*p* = 0.001	*p* = 0.002
Rich	1	1	1	1
Middle	0.64 (0.41–1.00)	0.67 (0.43–1.06)	0.67 (0.43–1.06)	0.68 (0.43–1.07)
Poor	0.41 (0.28–0.60)	0.48 (0.32–0.72)	0.47 (0.31–0.70)	0.47 (0.31–0.71)
Education differences	*p* = 0.001	*p* = 0.008	*p* = 0.007	*p* = 0.006
Both uneducated	1	1	1	1
Both equally educated	2.55 (1.48–4.39)	2.31 (1.32–4.05)	2.31 (1.33–4.02)	2.33 (1.33–4.06)
Husband better educated	1.32 (0.82–2.13)	1.27 (0.78–2.06)	1.25 (0.77–2.03)	1.26 (0.77–2.05)
Wife better educated	1.83 (0.97–3.48)	1.68 (0.88–3.23)	1.68 (0.89–3.14)	1.66 (0.88–3.13)
No of living children	*p* < 0.001	*p* < 0.001	*p* < 0.001	*p* < 0.001
3 or more children	1	1	1	1
2 or fewer children	2.43 (1.68–3.52)	2.03 (1.39–2.96)	2.10 (1.44–3.08)	2.10 (1.43–3.08)
**Women’s autonomy related**				
Exposure to media		*p* = 0.013	*p* = 0.017	*p* = 0.016
No exposure		1	1	1
Exposure		1.52 (1.09–2.12)	1.50 (1.07–2.09)	1.50 (1.08–2.10)
Pregnancy intentions		*p* = 0.011	*p* = 0.016	*p* = 0.014
Unintended		1	1	1
Intended		2.02 (1.17–3.46)	1.96 (1.13–3.40)	1.97 (1.14–3.10)
**Violence related**				
Partner control behavior			*p* = 0.029	*p* = 0.029
No behavior			1	1
1–2 behavior			0.64 (0.45–0.92)	0.63 (0.44–0.89)
3–5 behavior			1.17 (0.64–2.13)	1.01 (0.58–1.77)
Physical IPV			*p* = 0.836	NA
No			1	
Yes			0.94 (0.56–1.58)	
Sexual IPV			*p* = 0.519	NA
No			1	
Yes			0.80 (0.40–1.57)	
Emotional IPV			*p* = 0.485	NA
No			1	
Yes			0.82 (0.47–1.42)	
Any IPV			NA	*p* = 0.592
No				1
Yes				0.89 (0.58–1.35)
Nagelkerke’s R-square	0.211	0.227	0.239	0.237

**Model 1:** Age group, ethnicity, province, household wealth status, and witnessing parental violence. **Model 2:** Husband/Partner education, husband/partner alcohol use, women afraid of husband, marital control behavior displayed by the husband. **Model 3:** Education of women, exposure to media, women’s cash earnings, ownership of property, attitude towards the autonomy of sexual rights, and attitude towards wife beating. 1—reference category, *p* = *p*-value of the variables obtained from the test of model effects, IPV: Intimate Partner Violence. ANC: Antenatal Care, AOR: Adjusted Odds Ratio, CI: Confidence Interval. All values are weighted for the multi-stage sampling, cluster weight, and sampling weight.

**Table 5 ijerph-16-00895-t005:** Factors associated with institutional delivery among women in Nepal, NDHS 2016.

Characteristics	Institutional Delivery; AOR (95% CI)				
	Model 1	Model 2	Model 3	Model 4	Alternative Model
**Socio-demographic characteristics**					
Age group (years)	*p* = 0.155	*p* = 0.139	*p* = 0.162	*p* = 0.121	*p* = 0.107
15–24	1	1	1	1	1
25–34	0.87 (0.63–1.19)	0.84 (0.60–1.16)	0.85 (0.61–1.17)	0.84 (0.60–1.18)	0.83 (0.59–1.16)
35–49	1.32 (0.80–2.20)	1.29 (0.77–2.13)	1.29 (0.77–2.15)	1.34 (0.76–2.35)	1.34 (0.76–2.35)
Caste/Ethnicity	*p* = 0.001	*p* < 0.001	*p* = 0.002	*p* = 0.011	*p* = 0.007
Other castes	1	1	1	1	1
Brahmin/Chhetri	2.79 (1.70–4.57)	2.85 (1.73–4.70)	2.54 (1.54–4.16)	2.26 (1.37–3.72)	2.36 (1.43–3.91)
Janajati (Indigenous)	1.73 (1.07–2.77)	1.70 (1.05–2.75)	1.58 (0.96–2.59)	1.51 (0.92–2.49)	1.55 (0.94–2.56)
Dalit	1.94 (1.12–3.37)	1.94 (1.11–3.38)	1.83 (1.05–3.20)	1.68 (0.95–2.96)	1.70 (0.97–3.00)
Place of residence	*p* = 0.009	*p* = 0.010	*p* = 0.008	*p* = 0.031	*p* = 0.033
Urban	1	1	1	1	1
Rural	0.64 (0.45–0.89)	0.64 (0.45–0.90)	0.63 (0.45–0.88)	0.67 (0.47–0.96)	0.68 (0.48–0.96)
Household wealth status	*p* < 0.001	*p* < 0.001	*p* < 0.001	*p* < 0.001	*p* < 0.001
Rich	1	1	1	1	1
Middle	0.62 (0.42–0.94)	0.66 (0.43–1.00)	0.68 (0.45–1.03)	0.72 (0.47–1.12)	0.71 (0.46–1.11)
Poor	0.19 (0.13–0.28)	0.22 (0.15–0.34)	0.23 (0.16–0.35)	0.26 (0.17–0.39)	0.25 (0.17–0.38)
Education differences	*p* = 0.007	*p* = 0.013	*p* = 0.016	*p* = 0.082	*p* = 0.077
Both uneducated	1	1	1	1	1
Both equally educated	1.90 (1.10–3.26)	1.98 (1.14–3.41)	1.94 (1.10–3.39)	1.62 (0.96–2.75)	1.63 (0.97–2.76)
Husband better educated	1.22 (0.71–2.09)	1.33 (0.77–2.27)	1.31 (0.75–2.27)	1.21 (0.72–2.04)	1.22 (0.73–2.05)
Wife better educated	2.09 (1.12–3.91)	2.16 (1.15–4.05)	2.17 (1.14–4.12)	1.95 (1.04–3.65)	1.94 (1.04–3.62)
No of living children	*p* < 0.001	*p* < 0.001	*p* < 0.001	*p* < 0.001	*p* < 0.001
3 or more children	1	1	1	1	1
2 or fewer children	2.85 (2.07–3.94)	2.86 (2.00–4.10)	2.82 (1.98–4.03)	2.51 (1.70–3.69)	2.52 (1.71–3.71)
**Women’s autonomy related**					
Employment		*p* = 0.017	*p* = 0.007	*p* = 0.001	*p* = 0.002
Employed for cash		1	1	1	1
Unemployed		0.79 (0.54–1.17)	0.77 (0.52–1.14)	0.80 (0.54–1.18)	0.80 (0.53–1.20)
Employed, not for cash		0.61 (0.43–0.86)	0.58 (0.41–0.82)	0.53 (0.38–0.75)	0.54 (0.38–0.77)
Exposure to media		*p* = 0.454	*p* = 0.448	*p* = 0.864	*p* = 0.881
No exposure		1	1	1	1
Exposure		1.13 (0.82–1.55)	1.13 (0.81–1.57)	1.03 (0.73–1.44)	1.02 (0.73–1.43)
Pregnancy intentions		*p* = 0.921	*p* = 0.855	*p*= 0.402	*p* = 0.411
Unintended		1	1	1	1
Intended		0.97 (0.56–1.68)	0.95 (0.56–1.61)	0.78 (0.45–1.37)	0.79 (0.45–1.38)
**Violence related**					
Physical IPV			*p* = 0.026	*p* = 0.030	NA
No			1	1	
Yes			0.65 (0.44–0.95)	0.67 (0.47–0.96)	
Any IPV			NA	NA	*p* = 0.233
No					1
Yes					0.79 (0.55–1.15)
**Maternal Health Service Use**					
Sufficient ANC visits				*p* < 0.001	*p* < 0.001
No (<4 visits)				1	1
Yes (≥4 visits)				3.06 (2.20–4.26)	3.07 (2.21–4.26)
Nagelkerke’s R-square	0.299	0.306	0.311	0.358	0.355

**Model 1**: Age group, ethnicity, province, household wealth status, and witnessing parental violence. **Model 2**: Husband/Partner education, husband/partner alcohol use, women afraid of husband, marital control behavior displayed by the husband. **Model 3**: Education of women, exposure to media, women’s cash earnings, ownership of property, attitude towards the autonomy of sexual rights, and attitude towards wife beating. 1—reference category, *p* = *p*-value of the variables obtained from the test of model effects, IPV: Intimate Partner Violence. ANC: Antenatal Care, AOR: Adjusted Odds Ratio, CI: Confidence Interval. All values are weighted for the multi-stage sampling, cluster weight, and sampling weight.

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
