# Peer review of "The Role of Women’s Autonomy and Experience of Intimate Partner Violence as a Predictor of Maternal Healthcare Service Utilization in Nepal"

_ijerph, 2019, doi:10.3390/ijerph16050895_

Round 1

Reviewer 1 Report

A brief summary

This article presents the findings from the national, cross-sectional Nepal Demographic and Health Survey 2016. A range of secondary data analysis strategies were used to explore the factors that might relate to intimate partner violence against women.  The findings show that ANC is utilised more than institutional delivery and if a woman uses ANC she is also more likely to deliver in a hospital.  The findings also show that there are a number of social factors that influence access to maternal health services and that where women experience any IPV they will access health services less. 

Broad comments

A very important topic and one that needs greater attention, including to what solutions might be.  Overall, the article could be improved with better use of summaries, paragraphs and general editing.  The article would also benefit from more clearly stating what the main findings are, their implications including elaborating on potential policy levers there might be to address the gender relations that are discussed.  Currently there is a huge amount of detail within the article which means that the main points are lost.

Specific comments

Line 29 this is a very long sentence that needs to be broken up to make the points more clearly.

Line 38-39  what is the relevance of ‘delivery care’ and PNC for the rest of the article?

Para beginning line 37 Paragraph could be broken up to make the points more clearly. 

Para Line 63  Make this sentence more clearly about the reasons that women are not accessing care

Line 68   what is decision-making autonomy and what is economic autonomy?

Line 70   sentence is unclear

Section beginning Line 117 Tidy up the presentation of this section – maybe a Box is needed?

Results Section  Both the Tables and the explanation of the results could be made clearer.  Think carefully about what you need to show in your Tables to make the points you want to make.  Also, perhaps headings would be useful as you discuss the main issues in the results.

Page 9  Begin each of these sections with the main point before discussing the detail.

Discussion  Some very interesting points are made that would benefit from some clearer discussion of the main points you think that this analysis has uncovered.  Also informative headings throughout would be helpful.

Para beginning Line 333 why do you think ANC rates are higher than hospital delivery?

Line 347 there is a lot of research on geography and access, including from Pakistan and India as well as other countries such as New Zealand and Australia in relations to remote, communities.

Line 350  Is there a relationship between number of children and ses?

Line 360-363 these two sentences needs greater clarification.

Line 371-372  do you mean an association rather than an affect?

Limitation section  do you think the survey had any selection of response bias?  Especially

Line 427  Can you point to any possible policy levers that these findings suggest could be helpful?

Author Response

Dear editor, 

Please find the author response to reviewer 1 as an attached file.

Thanking you.

Reviewer 2 Report

This is a very interesting study. Furthermore, there is very little research with Nepalese population so the study is more novel and interesting. 

There are some recommendations to improve the manuscript: 

Abbreviations are included before they have been introduced. Authors should change it.

The objective of the study should be highlighted. In the abstract authors say that  This cross-sectional study aims to identify the relationship of women’s autonomy and IPV with maternal healthcare service utilization among married women of reproductive age in Nepal. However, this is not reflected in the theoretical framework nor in the methods or results. 

I recommend to make a deep theoretical revision about women’s autonomy. The information provided is really scarce. 

Finally, the discussion is very descriptive. I think they need more depth. 

There are some mistakes about format. Authors should revise the format.

Author Response

Dear editor,

Please find the author response to reviewer 2 as an attached file.

Thanking you.
